# Reducing Hospital Readmissions in Chronic Obstructive Pulmonary Disease Patients: Current Treatments and Preventive Strategies

**DOI:** 10.3390/medicina61010097

**Published:** 2025-01-09

**Authors:** Claudia Di Chiara, Giulia Sartori, Alberto Fantin, Nadia Castaldo, Ernesto Crisafulli

**Affiliations:** 1Respiratory Medicine Unit, Department of Medicine, University of Verona and Azienda Ospedaliera Universitaria Integrata of Verona, 37134 Verona, Italy; claudiadichiara95@gmail.com (C.D.C.); giulia.sartori.verona@gmail.com (G.S.);; 2Department of Pulmonology, S. Maria della Misericordia University Hospital, 33100 Udine, Italy; nadiacastaldo.nc@gmail.com

**Keywords:** exacerbation of COPD, hospitalisation, readmission, treatments, outcomes, strategies, length of hospital stay

## Abstract

COPD is one of the leading causes of death worldwide, so it represents a significant public health challenge. Over the years, new effective therapies have been proposed. However, the burden of COPD is still conditioned by frequent acute events defined as exacerbations (exacerbation of COPD-ECOPD), which have a significant impact not only on the patient’s quality of life but also on the progression of the disease, morbidity, and mortality. Related to the severity of the condition, ECOPD may require hospital admission and often repeatedly more admissions (readmission). The phenomenon of readmissions is a significant problem, contributing substantially to the utilisation of healthcare resources and the economic burden of COPD. Related contributing factors are still poorly understood, and managing the patients readmitted to the hospital with ECOPD may be challenging. Hospital readmissions should be optimally managed, including supporting and preventive strategies. Although early readmissions (30 days from discharge) are a marker of the quality of the patient’s care, we need to consider COPD patients globally. It is not sufficient to address just the acute events, so multidimensional management is necessary, able to follow the patient over time to identify, by a personalised approach, the correct treatment during and post hospitalisation and intercept any factor affecting the natural history of the disease, comprising the risk of hospital readmissions. In the context of the literature concerning respiratory medicine, particularly COPD patients, our narrative review analyses recent evidence regarding the current management of COPD hospital readmissions, aiming to propose preventive strategies helpful in clinical practice. The proposed strategies can potentially improve clinical outcomes and reduce healthcare costs when effectively implemented in practice.

## 1. Hospital Readmission: How and Why We Have This Problem

During the natural history of chronic obstructive pulmonary disease (COPD), patients may experience an acute worsening of the clinical condition and the daily symptoms, interrupting clinical stability, described in clinical practice as COPD exacerbation (ECOPD) [1]. ECOPD occurs when a patient experiences an acute worsening of respiratory symptoms due to a variety of factors, including bacterial or viral infection, as well as environmental or immunological causes [2,3,4]. The occurrence of ECOPD, described historically as a worsening dyspnea, increase in cough and sputum production, and requiring additional therapy, is accompanied by a rapid decline in health status, high risk of mortality, or other adverse outcomes such as intensive care unit (ICU) admission [5]. Although the definition of ECOPD is based on the onset of symptoms, or although additional therapy use may be helpful in clinical practice, this may conceptually oversimplify a more complex phenomenon [5] because there is no objective assessment of reported symptom worsening. In addition, this approach does not differentiate ECOPD from other most common causes of acute worsening in COPD, such as comorbidities (congestive heart failure or pneumonic exacerbation) [5,6,7]. A recent document [8] about the ECOPD severity of patients requiring hospitalisation classifies respiratory acidosis as the worst grade of ECOPD; in these patients, respiratory acidosis is proved to have a higher mortality rate in a short and intermediate period [9].

The rates of hospitalisation following ECOPD are unacceptably high [10] and are associated with several adverse patient outcomes, as well as contributing substantially to the healthcare resource utilisation and the economic burden of COPD [11]. For this reason, integrated management must be proposed, considering old and new approaches for hospitalised patients with ECOPD, to identify which factors may be associated with an increased risk of death [12] and which patients may have potentially an unfavourable clinical outcome. The presence of comorbidities, especially cardiovascular, may influence the mortality after hospitalisation for ECOPD [13]. In this context, a retrospective study considering autopsies suggests that common contributing causes of early death in patients hospitalised with severe ECOPD are concomitant complications such as heart failure (37%), pneumonia (28%), and pulmonary thromboembolism (21%) [14]. The Lung Health Study, a five-year multicentre trial that followed about six thousand smokers with mild COPD, showed that a single exacerbation may reduce the forced expiratory volume in the first second (FEV_1_) by 7 mL/year [15].

The phenomenon of relapse is most common in ECOPD. The need for a new hospitalisation (readmission), early if occurring in the 30 days from discharge, impacting significantly on a patient’s prognosis [16], has recently been considered a marker of the quality of the patient’s care [17]. Every effort should be taken to identify readmission predictors. Although several predictors have been identified [18], these do not fully recognise the phenomenon’s complexity. Therefore, managing patients readmitted with ECOPD may be a new challenge. Despite many interventions targeted at reducing readmissions, few strategies have been significantly effective. In this context, gaps that may be present in the existing literature may be related to the possibility of addressing the phenomenon of readmission individually, but not as an epiphenomenon of the complexity of severe COPD patients, and then globally. It is insufficient to address just the acute events: the reasons for readmission may be respiratory-related or not, so multidimensional management is necessary, following the patient over time, to identify, by a personalised approach, the correct treatment during and post hospitalisation and intercept any factor affecting the readmission. 

Our narrative review proposes pharmacological or non-pharmacological approaches and preventive strategies as suggestions (simple points of highlight) that are helpful in clinical practice.

## 2. Brief Methodology Used for the Review

We used the PubMed database for this narrative review, including English-language research studies (observational studies, clinical trials, systematic reviews, or narrative reviews) published from 1 January 2003 to 1 July 2024 interested in patients hospitalised with ECOPD. 

We adopted the search pattern of Medical Subject Headings (MeSHs) combined with accessible terms. The search strategies included the following terms: “chronic obstructive pulmonary disease”, “chronic obstructive airway disease”, “COPD”, “exacerbation”, “acute exacerbation”, “hospitalization”, “hospitalisation”, “hospitalized”, “hospitalised”, “readmission”, “outcomes”, “treatment”, “pharmacological”, “strategies”, and “predictors”.

One independent investigator (CD) identified specific records, completing a sequential review of all articles’ titles, abstracts, and full texts to identify eligible articles.

## 3. Why Do We Need to Highlight Hospital Readmission?

Readmission is a reasonable risk for patients surviving hospitalisation due to ECOPD [19,20], with different prevalences among countries [19,21]. In the US, around 19% of COPD patients are readmitted within 30 days, while in the UK, the rates are about 24% within 30 days and 43% within 90 days of discharge [10]. Data concerning the European COPD audit by Hartl et al. [19] showed that of more than fifteen thousand exacerbated and discharged COPD patients, approximately 35% were readmitted within 90 days and 20% within 30 days. These readmission rates have been relatively stable over the last decade, although differences in readmission rates may be associated with variations in care quality [19]. Of those readmitted patients, more than 50% of readmissions occur within the first 14 days following discharge from the hospital [22]. Most readmissions involve respiratory-related concerns, although only 27.6% are due to ECOPD [23].

In general, patients who are readmitted following a COPD hospitalisation are at greater risk of mortality and have worse outcomes relative to patients who are not readmitted [19]. Patients who are readmitted have a higher risk of death in comparison to those who are not readmitted (13% vs. 2.3%), and the patient’s age, the level of respiratory acidosis, the need for ventilator support, the Charlson index, and the number of previous admissions were considered as patient-related factors increasing the risk of readmission [19]. In a Spanish cohort study on almost four hundred ECOPD patients, the readmission 30 days from discharge for a new ECOPD progressively increases the mortality risk in periods of follow-up from six months to three years [16]. In the context of a meta-analysis of more than sixty-five thousand COPD patients, the cumulative readmission rates at 30, 90, and 365 days are, respectively, 7.1%, 12.6%, and 32.1%, with considerable variability between studies and countries [13]. In a multicentre study from Malaysia, the 90-day COPD readmission rate was 41% [21]; therefore, the readmission rates vary by country, with higher rates in Europe and the Western Pacific [24].

## 4. What Factors May Influence It?

The knowledge of risk factors for readmission in COPD has grown substantially, and identifying these is crucial for designing strategies to reduce their frequency and impact on patients [25]. We may generally consider patient-related and health-system factors [25]. Detailed characteristics of the studies considered and their principal outcome, including the readmission, are reported in Table 1.

### 4.1. Patient-Related Factors

Age determines short-term mortality in patients hospitalised for ECOPD [59]. A systematic review examining fifty-seven studies found that increasing age and male sex are the most prevalent readmission risk factors [25]. However, a recent meta-analysis found that younger COPD patients are more likely to be readmitted [29]. Although this may be contradictory, the severity of the condition requiring hospitalisation, which is also present in younger patients, may be the explanation. Another point of view is related to the interactions between age and payer status, in which younger patients with private insurance have a reduced risk of readmission [26,27,60]. Therefore, the insurance types of some countries increase (public healthcare coverage [27,61] or reduce (private care coverage) the risk of readmission. The geographical area of residence, marital status, and ethnicity are other readmission-related socioeconomic factors, with an increased risk for patients who live in less economically developed regions and do not have family support [60]. 

Concerning nutritional status, three malnutrition profiles may be identified in COPD patients: underweight with concomitant depletion of LBM (lean body mass) (60%), underweight with normal LBM, and patients of stable body weight with depletion of LBM [31,62]. In Spain, in three extensive retrospective studies with almost 800 thousand patients, malnutrition was associated with an increased risk of readmission within 30 days of discharge [32,63]. In general, malnutrition is associated with a 29% increase in 30-day readmission and a 73% increase in in-hospital mortality [33]. Patients with low BMI are more likely to be readmitted [35], while obese patients are less likely to be readmitted than non-obese patients [33].

Vascular and heart disease are the most common comorbidities observed in COPD, with a consistent impact on patient survival [36,37,64]. The pathophysiological mechanisms, which include endothelial dysfunction and coagulopathy, may increase susceptibility to exacerbations [65]. The systemic inflammation observed in COPD seems to be the determinant for pulmonary endothelial dysfunction and other factors, such as circulating endothelial progenitor cells, which appear to be in the pulmonary circulation of patients with COPD [65]. Furthermore, the tissue factor procoagulant activity and circulating levels of the thrombin–antithrombin complex are higher than in subjects without COPD, and this prothrombotic state increases the risk not only of stroke but also of clinical complications during severe exacerbations [38,41]. Pulmonary hypertension (PH), caused by endothelial dysfunction and pulmonary artery remodelling, has a significant prevalence in COPD patients (5% to 40%) [66]. The presence of PH in COPD worsens gas exchange and dyspnea, predisposes patients to right ventricular dysfunction and peripheral oedema, and is also associated with higher in-hospital mortality for patients readmitted for ECOPD [40].

The presence of osteoporosis increases the risk of vertebral fractures in COPD patients [39], impacting lung functionality [39]. Vertebral fractures can also lead to kyphosis, which restricts inhalation movements and reduces lung function parameters (9% reduction in forced vital capacity (FVC) per fracture) [39]. In the Evaluation of Obstructive Lung Disease and Osteoporosis (EOLO) study, the prevalence of vertebral fractures in COPD patients was 41% and correlated with COPD severity [67]. The physical approach to disease, as during an exercise retraining programme, may improve bone mineral density, muscle strength, and lung function, with indirect effects on exacerbations and readmission risk [42,68].

The severity of airflow limitation in COPD patients is a crucial risk factor for exacerbation requiring hospitalisation: a lower lung function in terms of post-bronchodilator FEV_1_/FVC ratio at baseline and post discharge was associated with an increased risk of readmission [69]. FEV_1_, in particular, has been found in some studies to be a predictor of hospital admissions for ECOPD, with a strong impact in predicting mortality and poor overall clinical outcomes, such as readmission after 30–90 days, intubation, and intensification of drug therapy [70]. In a retrospective study of COPD outpatients, the severity of FEV_1_ was a significant predictor for an increased risk of hospitalisations, as was the presence of significant comorbidities, such as diabetes or ischemic heart disease [71]. Few studies have studied the impact of COPD duration on the risk of readmissions. However, patients with a more extended history of COPD (>5 years) are approximately twice as likely to have frequent readmissions for ECOPD [70]. More data are available about the degree of dyspnea, which is reported to be associated with an increased risk of exacerbation and hospital readmissions [70].

In COPD patients, the dosage of biomarkers may help identify disease severity and progression, such as for urotensin-II (U-II) and transforming growth factor-β (TGF-β), playing a central role in the development and progression of fibrosis [72]. On the contrary, in the context of ECOPD, biomarkers may help identify patients at increased risk of exacerbation, with a different, disease-specific, early inflammatory response to infections [73]. In this context, fibrinogen and C-reactive protein (CRP) have been recognised as helpful biomarkers to discriminate pneumonic exacerbation from severe ECOPD [7]. CRP at discharge, associated with diabetes mellitus and one or more hospitalisations previously, were determinants of early readmission in hospitalised ECOPD [73]. In the context of biomarkers, more recently, the use of the blood eosinophil count has found a role as a predictor of the risk of exacerbation [74].

Regarding the role of infection, the majority of COPD readmissions for severe ECOPD are precipitated by respiratory tract infections, either viral or bacterial [43,44,75]. In severe ECOPD patients, a bronchial microbiological pattern corresponding to community-acquired pathogens (*Streptococcus pneumoniae*, *Haemophilus influenzae*, and *Moraxella catarrhalis*) is present in about 56% of positive samples, even though a high percentage of positive samples (44%) contained Gram-negative enteric bacilli (GNEB), *Pseudomonas*, and *Stenotrophomonas* [76]. An increased risk of exacerbation is associated with isolating new strains of bacteria, such as *Moraxella catarrhalis*, in the airways of COPD patients [45,46,77]. 

### 4.2. Health-System Factors

Factors related to this aspect may be associated with previous hospitalisations, discharge timing, and follow-up. Previous hospitalisations are essential in determining readmission risk; this phenomenon is well described in the literature by the definition of frequent exacerbation phenotype [78]. Patients with frequent exacerbations may have more comorbidities and severe disease severity, and hospitalisation in the previous year seems to be a common risk factor within 30- and 90-day readmissions after discharge [24]. Also, the need for an extended length of hospital stay (LHS), often related to a more severe ECOPD, has been reported as a risk of readmission [79]. In particular, patients with LHS ≥ 7 days identified a typology of severe ECOPD with common chronic baseline characteristics, including worse disease staging, worse symptom perception, long-term oxygen therapy use, and colonisation by *Pseudomonas aeruginosa* or microorganisms resistant to conventional treatment (MRCTs) [79]. Compared with ECOPD patients with normal LHS, a prolonged LHS increases the probability of readmission, especially for one readmission ≤ 30 days after discharge [79]. Similarly, the time to first readmission was lower in patients with prolonged LHS than in normal LHS (median 47 days vs. 95 days, *p* = 0.013) [79].

In this context, an inadequate or early discharge may worsen patients’ outcomes [24], suggesting that the post-discharge phase may impact the risk of readmission [30,47,80]. A recent article evaluated the impact of a pulmonologist follow-up visit during the month after discharge and reported that not attending the follow-up visit was associated with a significantly increased risk of readmission within 90 days of discharge [28,47,52]. Finally, in patients with moderate-to-severe ECOPD needing hospitalisation, in-hospital treatment failure was associated with a higher prevalence of readmitted patients in a period of 30 and 90 days and at 1 year [81]. Of note, related to the severity of ECOPD, the occurrence in a period of up to 7 days of at least one of the following conditions such as the need for non-invasive mechanical ventilation (NIMV) or ICU admission, the clinical persistence of signs of infection, and deaths from any causes defines the in-hospital treatment failure [81].

By graphical cartoons, Figure 1 and Figure 2 depict the identity of patients readmitted to the hospital and the factors related to this, respectively.

## 5. Reducing COPD Readmission: A Difficult Promise to Keep

Decreasing readmissions among patients discharged after being hospitalised for ECOPD is a worthy goal that could improve the patient’s health and save resources. Although the literature is full of articles investigating the phenomenon of readmissions and the related factors, there needs to be precise data on what prevents readmissions following COPD discharge, leaving pulmonologists needing clear guidelines regarding approaches to treating and reducing readmissions for these patients. Although 30-day readmission rates among patients hospitalised for ECOPD are of particular interest given the potential financial penalties imposed, there are few clinical trials designed to address the effects of interventions to reduce this outcome [51]. Also, the GOLD document suggests using an almost standardised approach for the COPD patient with a severe exacerbation who has just been admitted [2], and little data consider the possibility of intercepting the readmission phenomenon. Detailed characteristics of strategies or interventions related to the readmission outcome are reported in Table 2.

## 6. In-Hospital or Post-Discharge Pharmacological and Non-Pharmacological Strategies

Short-acting β-agonists (SABAs) [82,83] and short-acting muscarinic antagonists (SAMAs) are standard therapeutic practises, although no solid evidence supports this. In addition, there are no data on the use of dual bronchodilation [94]. More pieces of evidence are available about the use of steroids improving in-hospital (symptoms, LHS, treatment failure) and post-discharge (risk of relapse) outcomes. However, in COPD patients, the mechanism of resistance to steroids has been well documented and may represent a limit to their use and effectiveness [95]. According to the GOLD document, antibiotics are recommended for patients with moderate to severe ECOPD [2]. Benefits include a high rate of clinical cure within 30 days. However, mixed results have been reported regarding the impact of antibiotics on LHS, median time to next exacerbation, mortality, and overall treatment success [84,85]. However, no firm evidence exists about the antibiotic choice and treatment duration. Severe ECOPD patients have complex bronchial microbiological patterns and often interesting bacteria such as *Streptococcus Pneumoniae*, *Haemophilus influenzae*, and *Moraxella Catarrhalis*. Extended LHS can modify the microbial environment of these patients, making therapies ineffective due to the onset of resistance mechanisms [96]. In particular, it is proved that severe ECOPD patients with MRCT may have a longer LHS [96]. Frequent ECOPD patients have a high probability of developing pulmonary dysbiosis, which explains the difficulty in approaching the treatment [7]. Despite all, antibiotic therapy is a critical part of the therapeutic strategies for severe ECOPD, and this is particularly more evident in patients admitted to the ICU, in which they reduce treatment failures up to 4 weeks, all-cause mortality, and LHS [97]. In a recent study of about three hundred and twenty-five subjects hospitalised due to ECOPD [98], the effect of the classes and the association between antibiotic therapies on readmission rates was evaluated. Patients were grouped based on the type of antibiotic therapy received, according to guideline appropriation, non-appropriation, or no antibiotic therapy; in patients categorised as having received non-appropriation, the most common antibiotic choice was monotherapy with a fluoroquinolone-based regimen (levofloxacin). As a result, the treatment of ECOPD with antibiotics did not impact readmission rates, such as LHS, in-hospital mortality, or time to next exacerbation [98].

Among non-pharmacological treatments, we consider NIMV and pulmonary rehabilitation (PR). In hospitalised ECOPD with respiratory acidosis, the use of NIMV is strongly recommended with a high level of evidence [99]. Specifically, using NIMV is vital to prevent the need for endotracheal intubation and invasive mechanical ventilation in patients with mild to severe acidosis and respiratory distress, avoiding worsening respiratory functions during ECOPD [99]. The choice of the correct interface during NIMV may be a crucial variable for the success of this treatment [87,100]. In clinical practice, the helmet represents a valid alternative to increase the patient’s compliance, and the sequential use of a facial mask and helmet may produce a good tolerance from the patient and reduce the incidence of failure [101]. 

In COPD patients with persistent hypercapnia following a life-threatening ECOPD, a recent randomised trial has demonstrated that an improvement in time to readmission was observed when home NIMV was added to home oxygen therapy [92]. Despite the delay in time to readmission or death with nocturnal ventilation and reduced exacerbation frequency, the same study observed only a modest effect on health-related quality of life with the addition of home NIMV [92]. This aspect could be related to the severity of the disease and the time of history of disease in which NIMV was introduced, demonstrating the need to intercept sooner patients who have persistent hypercapnia for 2 to 4 weeks after the resolution of respiratory acidemia requiring mechanical ventilation [92]. When added to home oxygen therapy, physiological mechanisms that underpin the effect of the home NIMV could explain the clinical benefit of reduced hospital readmission [92]. Previous physiological studies have shown that home NIMV in patients with severe COPD improves the ventilatory response to hypercapnia, which could be expected to act as a clinically relevant effect of treatment, allowing a more robust and adaptive response to the adverse physiological challenge of an acute arterial partial carbon dioxide pressure (PaCO_2_) increase during ECOPD [92]. Finally, imaging data suggest that high-pressure NIMV may contribute to airway remodelling and improve ventilation–perfusion matching [92]. 

PR is recognised as a core component of managing individuals with chronic respiratory disease, and there has been considerable progress in our knowledge of its efficacy and scope [89,102]. This treatment consists of a structured programme of exercise, self-management education, and support to improve the physical and psychological condition of people with COPD and promote health-enhancing behaviours [90,103]. Several randomised controlled trials and two meta-analyses have suggested that the early initiation of PR can reduce the risk of hospitalisation and death [104,105]. Based on these data, guidelines for treating patients with COPD [106] recommend initiating PR within 3 weeks of an exacerbation [102,105]. One large retrospective study compared hospital readmission rates 1 year before and after PR initiation, finding that participation was not associated with fewer exacerbations [107]. These data contrast with the latest UK COPD PR National Audit, which found that PR was associated with a lower risk of hospitalisation and time spent there. In addition, a large observational study found that pulmonary rehabilitation within 90 days of discharge was associated with a lower risk of all-cause and COPD-specific readmission at 1 year. Recent evidence also demonstrates the efficacy of a PR program for hospitalised ECOPD patients [108,109,110]; additional resistance training during hospitalisation can prevent muscle deterioration and sarcopenia by improving lower-limb muscle strength [109], balance, and exercise capacity [111]. The initiation of PR is associated with a lower risk of readmission due to COPD and a lower risk for other conditions: exercise and the social components of PR lead to general improvements in health [111]. Despite all, PR services are only sometimes commissioned, and better referral uptake rates for early outpatient PR need to be increased. Although there may be several contributing factors, it is essential to acknowledge that some patients have difficulty accessing PR for various reasons, including lack of availability, social isolation, costs, transportation difficulties, and feeling unwell. Home-based PR and tele-rehabilitation programmes may mitigate some of these barriers [90].

## 7. Post-Discharge Programmes

As hospital discharge is a susceptible time for a patient admitted to the hospital due to an ECOPD, healthcare providers need to work together even though it may be difficult due to the lack of clear and common approaches [112]. Discharged ECOPD patients should receive comprehensive follow-up by the primary healthcare team. Discharge planning should involve the principal actor (patient) and several healthcare professionals, coordinated as a multidisciplinary hospital or community team, especially the general practitioner [113]. Social support and domestic arrangements are critical aspects of discharge planning. However, in some healthcare systems, there is significant pressure to reduce the LHS, resulting in hasty discharge before the patient reaches absolute stability, with a subsequent increased risk of readmission. Therefore, discharge timing may be crucial for intercepting [48,91,93]. Related to this reason, the Australia Lung Foundation [114] has created a checklist to suggest criteria for patients’ readiness for discharge, reported for the following points: The patient should be in a clinically stable condition and have had no parenteral therapy for 24 h [57];Inhaled bronchodilators are required less than four-hourly [86];Oxygen delivery has ceased for 24 h (unless home oxygen is indicated) [92];If previously able, the patient is ambulating safely and independently and performing activities of daily living [88];The patient can eat and sleep without significant episodes of dyspnoea;The patient or caregiver understands and can administer medications;Follow-up and home care arrangements (e.g., home oxygen, home care, Meals on Wheels, community nurse, allied health, general practitioner, specialist) have been completed [50,52,53,115].

Most readmissions following an index admission for ECOPD occur within 14 days post discharge, and most are unrelated to a new ECOPD. Several programmes targeting different disease states, including COPD, have reported improved readmissions using early outpatient hospital follow-up [54,116]. Implementing a programme focused on early post-discharge follow-up would be relatively straightforward; however, many potential barriers exist. For pulmonologists, having patients gain access to the office within 7 or 14 days of discharge may not be realistic, due to many clinicians’ current outpatient load [53,117].

A discharge plan, including information about the disease, medication use (inhalation therapy or oxygen devices), and a plan for managing worsening symptoms should be provided to the patients. Discharge bundles consist of a short list of evidence-based practises optimising the outcomes of patients [49,58,116,118]. However, a systematic review of the effectiveness of the COPD discharge approach showed that discharge bundles for patients with COPD led to fewer readmissions but did not significantly improve mortality or quality of life (QoL) [113]. In this context, a limitation of COPD discharge bundles may be related to the variability in levels of implementation, which has been commonly observed across studies. Hospital readmission is associated with increased morbidity, and for this reason, it is crucial to ensure continuous care and optimise treatment [119]. Miravitlles et al. [120] developed a practical and globally implementable COPD hospital discharge protocol for patients being discharged following ECOPD; this protocol contains four evidence-based care bundle items concerning (a) smoking cessation and assessment of environmental exposures; (b) treatment optimisation; (c) PR contents; and (d) the continuity of care. 

Smoking cessation refers to an intervention to support the discontinuation/quitting of tobacco smoking, including vapour or heated tobacco products. The most effective smoking cessation strategy in patients with COPD consists of pharmacotherapy combined with behavioural counselling [121]. Little is known about the effect of smoking cessation on the risk of future hospital readmissions; in a prospective study, quitting smoking was associated with a significant reduction in the risk of hospital admission [122]. In general, smoking cessation is the most effective strategy to slow down the progression of COPD and to reduce mortality in approximately 50% of COPD patients [122]. 

Treatment optimisation involves reassessing the treatment plan for every hospitalised patient with COPD and ensuring that patients receive the most appropriate pharmacological and non-pharmacological treatment at the right time. Global and local recommendations suggest that patients are established on their optimal maintenance therapy at hospital discharge based on an individualised assessment of exacerbation risk [2]. However, achieving this objective in clinical practice is not easy, because, even after hospitalisation, many patients remain sub-optimally treated, as described in a retrospective study, in which 16.6% of patients had not received maintenance COPD treatment during 4 months following the date of their first hospitalisation [123].

A combination inhalation therapy with LABAs (long-acting β-agonists), LAMAs (long-acting muscarinic antagonists), and ICSs [124] (inhaled corticosteroids) (formally triple treatment) has been shown to prevent exacerbations, reducing mortality in moderate to very severe ECOPD compared with dual therapies (LABA plus LAMA). Solid evidence shows that the prompt initiation of triple therapy after discharge reduces future exacerbations and healthcare resource utilisation compared with delayed initiation [125]. However, we should not forget that patient adherence to therapy is essential to optimise COPD management. Non-adherence to treatment, which is common among COPD patients, is another significant problem not only for individual health outcomes and for the care relationships between patients and healthcare professionals but also for the obvious economic implications. What the patient understands about using medications should also be assessed at discharge, and, where possible, educational initiatives should be implemented. Education is generally recommended in the standard position statements in diagnosing and managing COPD [55]. A better understanding of the disease and its management improves treatment adherence and reduces healthcare utilisation [120]. With the aim to optimise communication and cultural differences, educational materials should be adapted to individual needs; these materials may include written or pictorial information, videos, personalised written information on medication regimens, and teaching self-management skills [120]. 

Although many approaches have been systematically evaluated to improve outcomes, few interventions have consistently been associated with reduced readmissions for patients with ECOPD. COPD is a complex patient condition that can be followed over time. Often, all the preventive strategies aimed at reducing readmissions are poorly applicable due to a series of problems [55], such as poor domiciliary support, lack of understanding of post-discharge recommendations, and difficulty on the part of the health system [56] in organising the planned follow-ups within the pre-established time frame [88,126] Therefore, telemedicine visits are often proposed. However, telephone follow-ups usually leave the patient dissatisfied, contributing to reducing compliance with the care [118].

## 8. Bullet Points Summarising Practical Recommendations

Identify patients at risk for readmission by patient-related or health-system-related factors;Optimise pharmacological treatment during hospitalisation in order to also begin an early combined approach (LABA/LAMA/ICS);At discharge, evaluate the need for a specific treatment (smoking cessation, antibiotics, oxygen therapy, NIMV, PR);Consider a post-discharge plan programme (by telemonitoring visit);Consider the integration of more approaches in order to personalise the management.

## 9. What Hypothetically Needs to Be Done in the Future? A Short Perspective in Four Points

Consider readmission a multidimensional problem as an integrated part of multiple crucial factors in COPD patients, such as quality of life, social determinants, lack of compliance, and multimorbidity. Therefore, interventions aimed at reducing readmissions may need to go well beyond the only focus of COPD treatments to include improved patient education and behaviour modification through improved care pathways.Implement COPD knowledge in readmission content. Further efforts are needed to address some aspects related to the problem, such as the multimorbidity management of the readmitted patient. Recommendations and common approaches are not enough; we must also look at all the determinants of the patient’s global health.Increase the rigorous and scrupulous projects. Programmes to follow hospitalised patients often need to be more comprehensive concerning the costs that the stringent application of these entails. The programmes must address the quality of care. The readmission at 30 days may not be variable to attention, but it may be helpful in adjusting this limit to assess the impact of follow-up programmes.Communication is truly the key to understanding the problem. A poor outcome is often caused by poor communication during hospitalisation or care transfer. The patient should be adequately informed about the importance of knowledge of the disease in each aspect, from treatment to follow-up visits to the consequences of inadequate management.

## 10. Considerations About Limitation

Our approach in this review is purely narrative, describing evidence in the context of the literature, and the results are related to this approach. Clearly, an analytic approach involving selected inclusion criteria for each intervention and evaluating any possible publication bias for each original piece of data could give us an objective measure of efficacy.

## 11. Conclusions

In COPD patients hospitalised for ECOPD, readmission is a complex phenomenon, with a higher economic burden. Related contributing factors are still poorly understood, and managing it may be challenging. Although we have reported pharmacological and non-pharmacological strategies, reducing the readmission rate in COPD patients may be a difficult promise to keep. The possibility of shared paths that consider the phenomenon’s complexity, as of the disease, would be desirable to truly personalise the treatment. Future studies are necessary to increase scientific evidence on the topic with the final aim of defining how to manage this problem through solid recommendations.

## Figures and Tables

**Figure 1 medicina-61-00097-f001:**
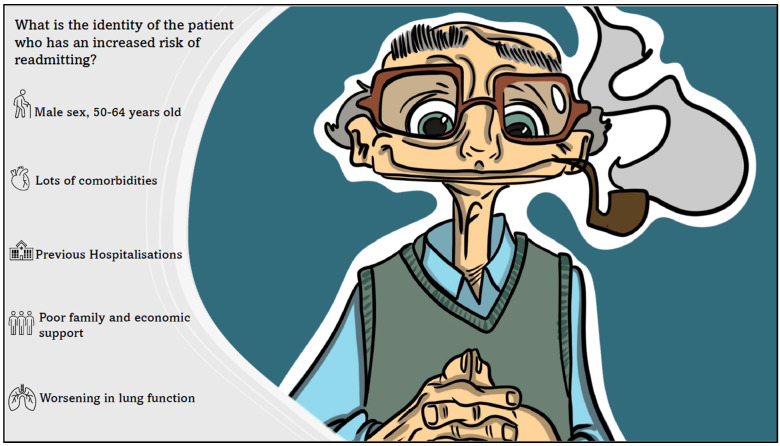
The identity of COPD patients readmitted to the hospital.

**Figure 2 medicina-61-00097-f002:**
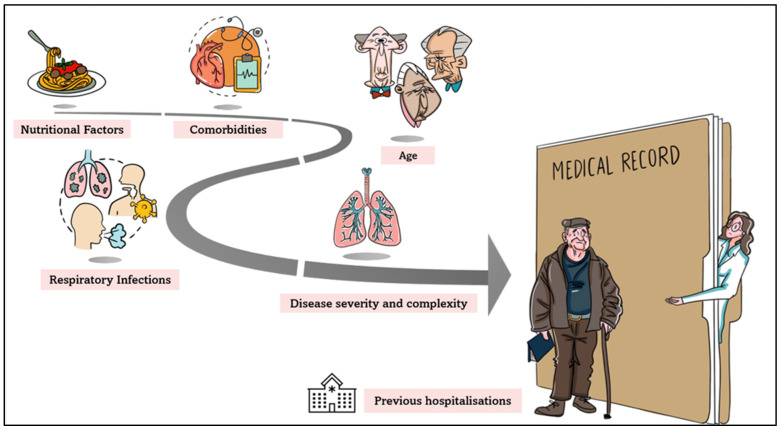
Factors related to hospital readmission.

**Table 1 medicina-61-00097-t001:** Characteristics of the studies considered and their principal outcome including the readmission.

Factors	Studies Considered	First Author	Year	Type of Study	Outcome			
					Readmission	Mortality	Economic Burden	Quality of Life
**Patient-related**								
Age and socioeconomic	6	Bhatt SP [26]	2017	OR	✔	✔	✔	
		Jiang X [27]	2018	OR	✔	✔		
		Swanson JO [28]	2018	OR	✔	✔	✔	
		Simmering JE [29]	2020	SR	✔	✔		✔
		Njoku CM [25]	2020	SR	✔	✔	✔	
		Myers LC [30]	2021	OR	✔			
Nutritional state	5	Celli B [31]	2004	OR	✔	✔		✔
		Vestbo J [32]	2006	OR	✔	✔	✔	
		Zapatero A [33]	2013	OR	✔	✔		✔
		Yu T [34]	2015	OR	✔	✔		
		Hunter LC [35]	2016	OR	✔	✔		
Comorbidities	8	Chambellan A [36]	2005	OR	✔	✔		
		Cote C [37]	2007	OR		✔		
		Brekke PH [38]	2008	SR	✔	✔		
		Graat-Verboom L [39]	2009	SR	✔	✔		
		Jyothula S [40]	2009	SR		✔		
		Vaidyula VR [41]	2009	SR		✔		
		Nuñez A [42]	2020	SR	✔	✔		
Respiratory infections	4	Leigh R [43]	2014	NR		✔		
		Wu X [44]	2014	SR	✔	✔		
		Wang Z [45]	2016	OR	✔	✔		
		Jang JG [46]	2021	OR		✔		
**Health-system-related**								
Telephone-based follow-up	2	Owens JM [47]	2015	SR	✔	✔		
		Bashir B [48]	2016	OR	✔	✔		✔
Physician follow-up	5	Sharma G [49]	2010	OR	✔	✔		
		Fidahussein SS [50]	2014	OR	✔	✔		
		Prieto-Centurion V [51]	2014	SR	✔	✔		
		Gavish R [52]	2015	OR	✔	✔	✔	
		Budde J [53]	2019	OR	✔			
Care pathways	5	Laverty AA [54]	2015	SR	✔	✔		
		Roche N [55]	2016	OR	✔			
		Vanhaecht K [56]	2016	OR	✔			
		Seys D [57]	2018	OR	✔			
		Shi M [58]	2018	SR	✔			✔

Abbreviations: OR, original research; SR, systematic review; NR, narrative review.

**Table 2 medicina-61-00097-t002:** Characteristics of strategies or interventions related to the outcome of readmission.

Type		First Author and Year	Effect on Readmission
Pharmacological strategies	SABA during admission	Bollu V 2013 [82]	30-day all-cause readmissions significantly lower in I vs. C (OR 0.69; 95% CI 0.51 to 0.92)
	LABA medication within 30 days of discharge	Bollu V 2017 [83]	All-cause readmissions are significantly lower in I vs. C (HR 0.53; 95% CI 0.30 to 0.96)
	Macrolide within 48 h of ICU admission	Kiser TK 2019 [84]	30-day all-cause readmissions significantly lower in I vs. C (OR 0.81; 95% CI 0.72 to 0.91); 30-day ECOPD readmissions not significant
	Azithromycin, initiated plus uploaded within 48 h of admission	Vermeersch K 2019 [85]	3-month respiratory readmissions significantly lower in I vs. C (RR 0.47; 95% CI 0.27 to 0.90)
	Use of a dry powder inhaler for LAMA treatment	Singer D 2020 [86]	COPD-related readmissions were significantly lower in I vs. C (OR 0.66; 95% CI 0.46 to 0.94)
Non-pharmacological strategies	Nocturnal NIMV at hospital	Struik FM 2014 [87]	1-year median respiratory readmissions were not different
	Session addressing core ECOPD risks: smoking cessation referral, GERD lifestyle modifications, anxiety and depressive symptoms, COPD education	Jennings JH 2015 [88]	Median time to ECOPD readmission is significantly shorter in I vs. C (10.5 days, IQR 3–15 vs. 18 days, IQR 11–28)
	Impact of PR after ECOPD on readmission risk in a real-world setting	Puhan MA 2016 [89]	Effects on hospital readmission are statistically significant in the meta-analysis but heterogeneous across trials
	Education, home-based exercise programme with follow-up (telephone calls regularly for 10 weeks)	Johnson-Warrington V 2016 [90]	30-day and 3-month respiratory readmissions were not significantly different in I vs. C
	Telephone follow-up (2 calls) focused on education, empowerment, and disease management; up to 30 days post discharge	Lavesen M 2016 [91]	30-day and 84-day all-cause readmissions not significantly different in I vs. C
	NIMV plus home oxygen therapy	Murphy PB 2017 [92]	1-year risk of all-cause readmission or death significantly lower in I vs. C (ARR 17.0%, 95% CI 0.1–34.0)
	Upon admission: standardised physician orders and care protocols, patient education Upon discharge: daily symptom monitoring, coordination of community care, COPD awareness day, PR programme	Agee J 2017 [93]	30-day all-cause COPD hospital admissions declined by 7.6% in I vs. C 30-day all-cause COPD readmissions declined by 46.03 % in I vs. C
Health system intervention	COPD discharge care bundle (based on national and international guidelines and input from other COPD programmes)	Laverty AA 2015 [54]	Pre–post analysis: in hospitals introducing discharge care bundle, readmission rates were rising before implementation and falling afterwards; readmissions within 28 days were +2.13% per year (pre-intervention) and −5.32% (post intervention)
	Hospitals with care pathway development and implementation, including (a) evaluation of organisation and quality of care; (b) providing a set of evidence-based key interventions; (c) training on how to develop and implement a care pathway	Seys D 2018 [57]	30-day COPD-related readmissions significantly lower in I vs. C

Abbreviations: SABA, Short-Acting Beta-Agonist; OR, original research; I, intervention; C, control group; OR, odds ratio; LABA, long-acting β-agonist; HR, hazard ratio; ICU, intensive care unit; RR, risk ratio; LAMA, long-acting muscarinic antagonist; NIMV, non-invasive mechanical ventilation; ECOPD, exacerbation of chronic obstructive pulmonary disease; GERD, gastroesophageal reflux disease; IQR, interquartile range; PR, pulmonary rehabilitation; AHRF, acute hypercapnic respiratory failure; ARR, absolute risk reduction.

## Data Availability

No new data were created or analysed in this study.

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
