# Peer review of "Reducing Hospital Readmissions in Chronic Obstructive Pulmonary Disease Patients: Current Treatments and Preventive Strategies"

_medicina, 2025, doi:10.3390/medicina61010097_

Round 1
Reviewer 1 Report
Comments and Suggestions for Authors
Dear author(s);
I have analyzed the manuscript entitled "Can we reduce hospital readmission in COPD patients? An updated narrative review on specific treatments and preventive strategies." The manuscript aligns with journal's scope as it: Focuses on the clinical management of a prevalent global health issue, COPD.Provides a comprehensive narrative review, analyzing treatment strategies and preventive approaches to reduce hospital readmissions.Includes multidisciplinary considerations, such as pharmacological and non-pharmacological strategies, which align with public health and clinical practice themes.
You can find my detailed comments below;
1) In the title; please consider highlighting "narrative review" more explicitly to immediately inform readers about the paper's methodological approach.
2) You can reduce expression slightly to improve accessibility for a broader audience in the abstract section.Also please provide a brief mention of the methodological framework (e.g., scope of literature reviewed) for added clarity.
3) The introduction establishes the study's relevance and the need for innovative management strategies.It provides a logical flow from general context to specific research objectives.Please include a more detailed breakdown of the gaps in existing literature to further justify the study's contribution.
4) The objective of the study is stated clearly but could be more concise. You have to revise the objectives to a single, focused statement for better impact.
5) The study uses a narrative review approach to synthesize existing evidence on COPD readmissions and related interventions. While the narrative review method is appropriate, there is no explicit mention of the databases searched, selection criteria, or analytic approach. So please add a brief methodology section detailing the search strategy, inclusion/exclusion criteria, and data synthesis process to enhance reproducibility.
6) Results are well-organized with supporting data from relevant studies. Tables summarizing study outcomes and strategies improve clarity. But You would provide more data visualizations (e.g., graphs) to enhance reader engagement and would incorporate quantitative summaries where applicable (e.g., effect sizes of interventions).
7) The discussion section integrates findings with broader literature effectively and acknowledges study limitations. The authors call for future research and improved care pathways. However You could expand on the limitations of the narrative review approach and potential publication bias. Please propose specific research questions or designs for future studies.
8) The conclusion section is succinct and align with the objectives. You could consider summarizing the practical recommendations as bullet points for clarity of this section.
9) References are formatted properly, but more details about the studies reviewed (e.g., in an appendix) could strengthen transparency. So please include a supplementary appendix summarizing key studies and their findings.
10) Please discuss potential biases and methodological constraints more comprehensively.
Reviewer 2 Report
Comments and Suggestions for Authors
Dear Authors,
It has been a pleasure to read your work. It is well organized and documented.
My comments are the following:
1. I miss a methods section - tell me the search strategy (which database you searched and when), the eligibility criteria for including the studies you mentioned (for example in Table 1) and data collection details (methods to confirm accuracy, piloted forms?, etc)
2. In the final section or in section 7, maybe mention which are you paper's limitations.
3. I must say the two figures are the only thing I did not like - all art is subjective, I know, but please reserve them for a poster to be shared to family practice to inform the patients? Also, putting them one after another is not really a good practice. Maybe replace them with a table? And add some percentages to those characteristics, like you did everywhere else in the paper? I think that would be more appropriate for this context.
4. From a formatting perspective I would've liked to see a way to visually highlight ideas in paragraphs (like bold / italics). I actually did underline the words on my copy of the paper to see the idea flow better. But I understand if you chose not to do this, as it is more traditional the way it is.
5. Minor inconsistencies:
line 36 - review the title for grammar - it sounds wrong to me
lines 51-53 - you mention one document but cite two. Rephrase.
line 184 - in this latter - sounds wrong to me
lines 471-476 - delete the non applicable parts, mention just what is applicable
Sorry for giving you a little more work over the winter holidays, hope it does not take much away from your time off. Best wishes!
Reviewer 3 Report
Comments and Suggestions for Authors
Title and Abstract:
1- Title: The current title is clear and descriptive. However, to emphasize the focus on "readmission," the title could be revised as follows: "Reducing Hospital Readmissions in COPD Patients: Current Treatments and Preventive Strategies." This revision would provide a more direct reflection of the study’s scope and purpose.
2- Abstract: The abstract summarizes the main points of the study effectively. However, it would benefit from a more specific concluding sentence about the practical implications of the proposed strategies. For instance: "The proposed strategies have the potential to improve clinical outcomes and reduce healthcare costs when effectively implemented in practice."
3- Introduction: The introduction provides a good background on the significance of COPD and the burden of readmissions. However, incorporating recent studies to highlight the economic and clinical impact of COPD readmissions could strengthen the context. For example, citing studies on global readmission trends would provide a broader perspective.
4- Methodology: The methodology section is not included in detail, but if the review is narrative, the criteria for selecting the studies should be explicitly stated. This includes search databases, keywords, and inclusion/exclusion criteria.
5- Content and Discussion: The manuscript effectively discusses various factors contributing to COPD readmissions and potential strategies. However:
Expand on the role of biomarkers, such as Urotensin-II and TGF-β, in identifying high-risk patients. Cite the relevant study:
Kilinc M, Demir I, Aydemir S, Gul R, Dokuyucu R. Elevated Urotensin-II and TGF-β Levels in COPD: Biomarkers of Fibrosis and Airway Remodeling in Smokers. Medicina (Kaunas). 2024 Oct 24;60(11):1750. doi: 10.3390/medicina60111750. PMID: 39596935; PMCID: PMC11596865.
Discuss the potential for integrating personalized medicine approaches in COPD management.
The figures included (if any) could be referenced more explicitly within the text to guide the reader.
6- References: The reference list is comprehensive, but the authors should ensure it includes the most recent and relevant studies. Ensure references align with the discussion, particularly regarding biomarkers and preventive strategies.
Verify and update older references, as COPD management has seen advancements in recent years.
7- Language and Style: The manuscript is well-written, but minor grammatical improvements are needed for clarity. For example:
Rephrase sentences to avoid passive voice where possible.
Simplify complex sentences for better readability.
8- Conclusion: The conclusion provides a concise summary but could include specific recommendations for future research directions, such as exploring the efficacy of novel interventions in reducing readmission rates.
General Recommendation:
With minor revisions, this manuscript has the potential to significantly contribute to the field of COPD management. It addresses a crucial issue with practical implications for reducing hospital readmissions. Including a broader range of recent literature and refining the language will enhance its overall quality.
